# More than the SRY: The Non-Coding Landscape of the Y Chromosome and Its Importance in Human Disease

**DOI:** 10.3390/ncrna10020021

**Published:** 2024-04-10

**Authors:** Emily S. Westemeier-Rice, Michael T. Winters, Travis W. Rawson, Ivan Martinez

**Affiliations:** 1West Virginia University Cancer Institute, West Virginia University School of Medicine, Morgantown, WV 26506, USA; ew0065@mix.wvu.edu; 2Department of Microbiology, Immunology and Cell Biology, West Virginia University School of Medicine, Morgantown, WV 26506, USA; mtwinters@mix.wvu.edu (M.T.W.); twr0001@mix.wvu.edu (T.W.R.)

**Keywords:** Y chromosome, non-coding RNAs, lncRNAs, miRNAs, circRNAs

## Abstract

Historically, the Y chromosome has presented challenges to classical methodology and philosophy of understanding the differences between males and females. A genetic unsolved puzzle, the Y chromosome was the last chromosome to be fully sequenced. With the advent of the Human Genome Project came a realization that the human genome is more than just genes encoding proteins, and an entire universe of RNA was discovered. This dark matter of biology and the black box surrounding the Y chromosome have collided over the last few years, as increasing numbers of non-coding RNAs have been identified across the length of the Y chromosome, many of which have played significant roles in disease. In this review, we will uncover what is known about the connections between the Y chromosome and the non-coding RNA universe that originates from it, particularly as it relates to long non-coding RNAs, microRNAs and circular RNAs.

## 1. Introduction

Throughout history, humans have been plagued with various diseases, from physical illnesses like cardiovascular infarctions and cancer to psychological illnesses such as depression and anxiety. Sex differences between many diseases are well documented, such as lung cancer rates, asthma, cardiovascular infarctions, and depression [1,2,3]. Additionally, there are notable differences in cancer mortality and survival rates between men and women, with men having an increased risk of developing and succumbing to cancer [4]. This sex disparity can also be found in relation to SARS-CoV-2 infection, with men more likely to die from infection, while females are more likely to have symptoms of long COVID [5,6,7]. In general, there are three major categories of factors that influence the disparity in disease occurrences between males and females. These are genetic, hormonal, and environmental.

From the genetic perspective, biological males have only one X chromosome and have an additional haploid chromosome, the Y chromosome. Early in genetic history, the Y chromosome was originally considered to be genetically inert, except for the SRY gene. The presence of this gene and its corresponding protein, the testes determining factor (TDF) [8], results in the presence of testosterone during development, creating the male gonads and sex organs. In contrast, biological females have two X chromosomes, which do not contain the SRY gene, resulting in the formation of female sex organs and an increase in estrogen production. From the hormonal standpoint, males and females differ in the concentration of the sex hormones: testosterone and estrogen/progesterone throughout the body. These hormones are regulated differently in biological males and females, dependent on their blood concentrations and site of production [9]. Females tend to have more estrogen and progesterone, while males have much higher levels of testosterone, leading to differences in endocrine diseases [10]. Estrogen plays an important role in immune function, and thus females are more likely to have autoimmune diseases and thyroid autoimmunity [11,12]. Environment is the third major category of factors that influences the differences in disease states. Contributing factors to these environmentally based differences include environmental pollution, climate change and employment requirements. Overall, women are more sensitive to environmental pollution and climate change, while men are more likely to receive workplace injuries and chemical or biological exposures [13,14,15]. Additionally, men are more likely to smoke and drink, which has contributed to their higher rates of cancer throughout history [16]. While each of these factors has an important impact on disease disparity, the impact of the genetic differences between males and females is generally overlooked in favor of the hormonal impact of estrogen.

It is important to note the difference between biological sex and gender in the scope of this paper. Because the focus of this review is on the Y chromosome specifically, we will be specifying biological sex as males with an XY chromosomal pair and females with an XX pair of sex chromosomes. The authors acknowledge the role that gender may play in some of these diseases as well, as gender influences upbringing environment, social interactions, and other societal pressures. This can also influence the prevalence of diseases due to environmental interactions. This remains an active area of research with many interesting avenues to investigate, but as we are specifically focusing on non-coding RNAs originating from the Y chromosome, this review will be focusing on the impact of the Y chromosome on biological sex throughout this article.

Non-coding RNAs, a burgeoning field, particularly including those coming from the Y chromosome, offers incredible potential on tackling these differences between males and females. Here, we present current literature on the non-coding landscape of the Y chromosome, with particular focus on long non-coding RNAs, microRNAs, and circular RNAs. The non-coding landscape holds significant potential for personalized medicine, with an emphasis on using non-coding RNAs to create more specific treatments. The advent of mRNA vaccines for treatment of SARS-CoV-2 has shown the efficacy of this type of nucleic acid treatment and its potential to help a wide range of diseases [17,18]. The methodology used to create this review was systemic searches through established databases such as PubMed and Google Scholar. Therefore, we have compiled the most recent literature on the Y chromosome and its non-coding landscape, creating a comprehensive study wherein the non-coding RNAs that have been characterized are presented with the intention of increasing visibility of the use of these non-coding RNAs in diagnostic, prognostic, and therapeutic models of sex disparities.

## 2. Y vs. X Chromosome

The third-smallest chromosome and containing the fewest number of protein-coding genes, the Y chromosome has just over 62 million base pairs [19,20]. Evolutionary pressure has shifted the Y chromosome from being comparable to its partner sex chromosome in gene number to its present state, with only 55 protein-coding genes present on the current Y chromosome [21]. The Y chromosome in humans, soon after its divergence from the X chromosome, lost most of its genes to the autosomes; however, it retained its likely most important gene: the sex determining gene, SRY. Interestingly, it has been shown that ancestrally, the genes retained on the Y chromosome were not random, but were evolutionarily selected [22]. Contrary to the human Y chromosome structure, the mouse Y chromosome has retained its evolutionary significance, being 99.9% euchromatin and holds the genetic code for nearly 700 genes (Figure 1) [23]. This has created a particular paradox for studying the non-coding landscape of the Y chromosome within in vivo studies. While many studies can use the mouse as a model organism, those focusing on the genetic impact of the Y chromosome have additional difficulty.

In contrast, the human X chromosome has over 150 million base pairs containing the genetic code for more than 1400 genes [20]. While this creates an expected gene imbalance between males and females, evolution has fueled the inactivation of the second X chromosome in females through a known mechanism: the Xist long non-coding RNA. Xist is an RNA encoded on the X chromosome that selectively interacts with the X inactivation center to silence transcription from the second X chromosome [24]. This process has been studied since the 1990s, when Xist was first identified and characterized [25]. Through this process, Xist is critical for creating genetic balance between males and females. However, there are between 15% and 50% of genes that can escape X chromosome inactivation, based on tissue type [26]. These genes include those in the pseudo-autosomal regions, genes with dosage sensitivity or immune-related, or they can even be bystanders near other escape genes [27]. Because many of the genes are immune related, it is likely that females are receiving a “double dose” of immune-related genes, thereby increasing their immune function. In turn, this may be impacting the genetic differences between males and females in response to certain disease states.

Initial attempts to sequence the human genome expected to identify hundreds of thousands of genes to verify the complexity of the human race. However, what they found was the presence of only 30–40,000 genes, the rest being considered “junk DNA” [28]. Shortly after, this “junk DNA” was found to be transcribed very frequently in the form of non-coding RNAs [29]. These new classes of non-coding RNAs challenged the central dogma of biology, in which DNA is transcribed to RNA and directly translated to protein, while simultaneously giving rise to a new avenue wherein the complexity of the human genome may begin to be explained [30]. This complexity can be seen through the changes in human genome annotations since the first sequencing of the human genome in 2004 [31]. However, initial sequencing of the Y chromosome presented the sequences only for the MSY regions, accounting for just over 50% of the entire chromosome [32]. The currently most used version of the human genome was started by the Human Genome Project in 2013. Additional annotations were provided with improvements in technology in 2019, named GRCh38 [33]. The Telomere-2-Telomere consortium released their most recent full sequence of the human genome in 2022, with complete annotations expected to be released over the coming years [34]. Early annotations of T2T-CHM13v2.0 have been released and are currently available through the National Center for Biotechnology Information (NCBI). To avoid additional complexities, they performed the sequencing on the female CHM13 cell line. The progression of sequencing of the Y chromosome has its own challenges, stemming from the long segments of heterochromatin and complex repeats within its genetic code, including palindromes, duplications of various segments, inversions and many tandem repeats [35,36,37,38]. One of the largest sections of the Y chromosome’s heterochromatic regions is the presence of human satellite III [39]. This section is characterized by the pentanucleotide repeat TTCCA, and while the human satellite III sequences occur on chromosome 1 and 9’s heterochromatin and chromosomes 13–15, 22, there is no cross-reaction between the chromosomes, indicating the sequences are unique, albeit interspersed with the TTCCA repeats [40,41]. Human Satellite III has also been shown to express various transcripts, specifically in response to stress, indicating its transcription potential, even though it is comprised of heterochromatin [42,43,44,45]. This theoretical and yet also very real enigma that surrounds the sequence of the human Y chromosome creates an adverse environment for identifying the genetic differences between males and females.

Late in 2022, a manuscript identifying the complete sequence of the human Y chromosome from the National Human Genome Research Institute (NHGRI) at the National Institutes of Health (NIH) in the United States was released as a preprint through BioRxiv [19]. In August 2023, the manuscript was released and published [19]. As with the full T2T genome sequence, full annotation is expected within the next few years. Additional research published concurrently with the full sequence of the Y chromosome was the full sequence of 43 variants of the human Y chromosome, identifying the incredible variation in the Y chromosome within the human population, across nearly 190,000 years of evolution [46]. The release of these sequences has been instrumental to beginning investigation into the Y chromosome’s mysterious black box. With the release of the Y chromosome’s sequence, work to understand the impact of the Y chromosome has only begun. As it is a haploid chromosome, and the third-smallest chromosome, men will begin to lose their Y chromosome as they age, which can then be accelerated by smoking [47,48]. This loss of the Y chromosome is incomplete across cell types and has thus been named mosaic loss of Y (mLOY), and has been associated with a higher rate of Alzheimer’s disease [49], cardiac fibrosis [50], incomplete immune response [51], and implicated in increased risk of developing cancer [52]. This risk has been seen across various cancer types, as seen by a recent publication by Muller et al. in 2023 that took a pan-cancer approach to identifying mLOY [53]. Mueller describes that increased genetic instability and aggressive mutation rates are associated with mLOY [53]. This work supports research done across cancer types, with a publication in 2019 showing worse prognosis relating to mLOY in head and neck squamous cell carcinoma [54]. With the relatively low number of genes associated with the Y chromosome, an important question remains as to how mLOY can impact mutation rates, and further research into the non-coding landscape of the Y chromosome may present the answers.

## 3. Non-Coding RNAs

Non-coding RNAs (ncRNAs) is a field has been blossoming throughout the last century. Ribosomal (rRNAs) and transfer RNAs (tRNAs), the most common classes of ncRNAs, were identified as early as the 1950s [55,56]. Newer classes of ncRNAs, such as long non-coding RNAs (lncRNAs), microRNAs (miRNAs) and circular RNAs (circRNAs) have been referred to as part of the dark matter of the human genome [57], as they generally originate from repetitive regions and transposable elements within the human genome, once thought to be junk DNA [58]. It was not until 1990 (lncRNAs) [59], 1993 (miRNAs) [60] and 1991 (circRNAs) [61] that each type of ncRNA was recognized as functional and their own fields began to grow. Non-coding RNAs can be broken down into two major classes: housekeeping and regulatory ncRNAs [62]. Housekeeping ncRNAs are ubiquitously expressed regardless of cell type and are important regulators of primary cellular functions, such as rRNAs, tRNAs, and small nuclear and small nucleolar RNAs. Regulatory ncRNAs are more restricted in their expression, usually tissue and developmental stage-specific. These ncRNAs are further broken down into two categories: long non-coding RNAs and small non-coding RNAs. Within these categories, long non-coding RNA counts both linear long non-coding RNAs and circular non-coding RNAs. Some of the most common small non-coding RNAs include microRNAs, piwi-interacting RNAs (piRNAs), silencing RNAs, and Y RNAs. The non-coding universe expands beyond those found within mammals, including repeat associated small interfering RNAs (rasiRNAs) found in plants and *Drosophila* [63,64]. Currently, there is no evidence of rasiRNA expression in human cells, including the Y chromosome, and Y-linked piRNA in human cells is not very well characterized [65,66,67]. For this reason, this review will focus on three major classes of non-coding RNAs—long non-coding RNAs, microRNAs, and circular RNAs—that are encoded on the Y chromosome.

NcRNAs are at the forefront of efforts to create personalized medicine. Not only from the research standpoint, but for policy and guideline creation, there must be standardized practices that protect the patient and their genetic code, while still using genetically based treatments. Many researchers and healthcare professionals have identified various ncRNAs of interest in diseases, such as endometrial and colorectal cancer [68,69,70]. Long non-coding RNAs have been identified as biomarkers in endometrial cancer, and Y RNAs have been slowly arising as likely targets for personalized medicine [68,71]. There have been numerous studies on delivery systems, particularly lipid nanoparticles due to the success of the COVID-19 vaccines [72,73,74]. The future of personalized medicine would benefit from additional insight on the Y chromosome, potentially by serving as a biomarker, and Y chromosome presence informing clinical treatment modalities.

### 3.1. Long Non-Coding RNAs from the Y Chromosome

Linear long non-coding RNAs (lncRNAs) are a class of ncRNAs that is among one of the most functionally and genetically diverse, and are highly prevalent on the Y chromosome, especially when compared to the numbers of other ncRNAs, such as other housekeeping or regulatory RNAs. lncRNAs have the highest versatility due to their reliance on their secondary and lncRNAs from the Y chromosome tertiary structures to function, rather than their sequence, which is important, as many lncRNAs are tissue-specific and are not evolutionarily conserved [75]. Additionally, most lncRNAs are low expressors under normal conditions and only increase in expression during disease states [76]. There are four categories of lncRNAs: antisense, sense, intergenic and intronic [77]. Antisense lncRNAs are named for the gene they are closest to, on the antisense DNA strand to the gene, tagged with a -AS, such as ASMTL-AS1 [78]. Sense lncRNAs are found within the same strand of DNA as the coding gene they are named for, and usually contain exonic regions that are not translated, such as DAPLAR [79]. The most common lncRNAs are the intergenic lncRNAs (also known as lincRNAs), which generally are not found near or overlapping any coding gene [80], such as the linc-SPRY3 family [81]. As evidenced by their name, intronic lncRNAs are lncRNAs that come from splicing events of intronic regions of coding genes. The Y chromosome has each type of lncRNA; however, with only 50 coding genes, intergenic are the most common lncRNAs present.

Transcribed by RNA polymerase II, many lncRNAs are processed similarly to mRNAs, containing a poly-A tail and a guanine cap. Nuclear retained lncRNAs generally act as pre- and post-transcriptional regulators. Pre-transcriptional processing is performed through many lncRNAs, such as Xist, a lncRNA that causes epigenetic changes, crucial for X chromosome inactivation [82], or MANTIS, which has been shown to influence angiogenic marker expression [83]. MALAT1 has been shown to perform post-transcriptional processing of mRNAs by modulating splicing and recruiting RNA-binding proteins [84,85]. Many lncRNAs will also localize to the cytoplasm, where they can work as competitive endogenous RNAs, like TUG1 [86] and TTTY15, increase stabilization of mRNAs, such as Sros1 and NEAT1 [87,88], or decay of mRNAs, like EPR and linc00525 [89,90], regulate protein localization, like ADEPTR [91], or control post-translational processing of proteins, like Cerox1 [92]. Previously, it was assumed that lncRNAs lacked a functional open reading frame to produce any coded peptides; however, recent advances in sequencing have identified many lncRNAs have the ability to code so-called micropeptides that contain fewer than 100 nucleotides [93,94]. With their diverse functions, lncRNAs are not unique to autosomes, with many being identified from the X and Y chromosomes as crucial for normal development, and their dysregulation has been implicated in many diseases. Lncipediav5.2 cites nearly 600 lncRNAs identified on the Y chromosome; however, many are speculatively placed on the database without validation [95]. Of the characterized lncRNAs, many have been identified to impact disease states, and many have specific counterparts found on the X chromosome. This double exposure further pressures the evolutionary change that drives the downsizing of the Y chromosome, though there are still some lncRNAs that are specific to the Y chromosome.

Over time, many lncRNAs have been identified to originate from the Y chromosome. These lncRNAs and their functions span from early development to disease onset, biomarkers for cancer and even cancer progression. Table 1 shows some of the characterized lncRNAs originating from the Y chromosome and their known functions [81,96,97,98,99,100,101,102,103,104,105,106,107,108]. There have been other lncRNAs identified to originate from the Y chromosome; however, many have not been characterized, such as the rest of the TTTY family. With the advances in sequencing over the most recent years, it is likely that emerging research will elucidate the functions of these previously unexplored lncRNAs.

### 3.2. MicroRNAs from the Y Chromosome

MicroRNAs are small ncRNAs, processed in segments of about 23 nucleotides in length [109]. In general, microRNAs are categorized into families based on their seed region. The seed region, which generally begins with the second 5′ nucleotide of the miRNA, is an eight-nucleotide region crucial for the binding of the miRNA to its target [110]. Criteria for identifying and annotating miRNAs were originally described by Ambros et al. and based around expression and biogenesis criteria. Proposed miRNAs had to agree with at least one criterion in both categories, due to similarities in expression to silencing RNAs (siRNAs) and a non-unique biogenesis pathway. Once determined to be a miRNA, the RNA was given a name “mir-” with its following number related to the order in which it was discovered [111]. Over time, the naming convention has shifted slightly, looking for homologues and locational significance. These families the miRNAs are categorized in are dependent on the sequence the seed regions target, and may or may not originate from chromosomal neighbors [109]. Understanding the role of specific miRNAs has been challenging. In humans and mice, most pathways have redundancy in their miRNA regulation, meaning specific knockdowns do not always create a phenotype [112].

The creation or biogenesis of human miRNAs can begin with several primary RNA substrates, but canonically begins with an intragenic, primary miRNA transcript (pri-miRNA) generated by RNA polymerase II/III from introns or exons within the nucleus of the cell [113,114,115]. The primary transcript is ~70 nt and consists of an RNA transcript with a short hairpin structure and terminal loop [116,117]. The pri-miRNA serves as a binding site for DiGeorge syndrome critical region 8 (DGCR8), which recognizes N6-methyladenyladensoine following METTL3 modification, and other nucleotide motifs at the primary miRNA hairpin structure, recruiting the ribonuclease III enzyme Drosha, forming the microprocessor complex, which cleaves the miRNA duplex at the base of this hairpin [113,118,119]. This newly created precursor miRNA is then transported to the cytoplasm via an exportin-5–Ran-GTPase complex [120]. Here the precursor is modified again by the Rnase III endonuclease Dicer, which cleaves the terminal loop and leaves a mature miRNA duplex, consisting of 5p and 3p strands—derived from the directionality of the 3′ or 5′ end of the miRNA hairpin [121]. Each of the strands within the duplex can be loaded into the Argonaut-2 (Ago-2) protein of the RNA-induced silencing complex (RISC) in an ATP-dependent fashion [122]. The loaded strand is known as the guide strand, and normally interacts with the 3′UTR of the target mRNA, leading to its degradation via decapping and deadenylation of the mRNA, which decreases mRNA stability and leads to translational repression [123].

The non-canonical biogenesis of miRNAs uses different combinations of proteins from the canonical pathway, and largely depends on the RNA substrate used. These can be divided into grouped pathways as Drosha/DGCR8-independent and Dicer-independent, with the substrate of each resembling that of the opposite, dependent proteins [115,124,125,126]. Drosha/DGCR8-independent precursor miRNAs resemble Dicer substrates—Mirtrons, from the splicing of introns—and are directly transported to the cytoplasm via exportin-1 for Dicer processing [115,126,127]. Dicer-independent precursor miRNAs are processed by Drosha from short hairpin (sh-RNA) transcripts, and as they are too short to act as Dicer substrates, the entire precursor miRNA is loaded into Ago-2, which slices the 3p strand, leading to 3′-5′ trimming of the 5p strand, which completes the maturation of the miRNA [124,128]. MiRNAs can also bind to other regions of their target mRNA, including the 5′UTR, coding sequence, and within promoter regions [129,130]. Binding to the 5′UTR and coding regions has silencing effects on gene expression, while binding at promoter regions induces transcription [130]. mRNA sensitivity to miRNA gene regulation largely depends on mRNA secondary structures, and alternative splicing can affect miRNA-binding sites [131,132].

mRNA sensitivity to miRNA gene regulation largely depends on mRNA secondary structures, and alternative splicing can affect miRNA binding sites [131,132]. This, however, is not the only function many human miRNAs carry, with some human miRNAs having many canonical functions and many having more than one primary function. Major roles of miRNA in the cell include but are not limited to cell–cell communication, proliferation and apoptosis activation or inhibition, and the epithelial–mesenchymal transition and invasion in the context of cancer [133,134,135,136,137].

In recent years, only four microRNAs have been found to be processed transcripts originating specifically from the Y chromosome [138,139,140,141,142,143,144,145,146,147,148] (Table 2). The biological function of some of these microRNAs has been found, but many must be investigated further. MiR9985 has been found to be differentially expressed in the skeletal muscle transcriptome in response to burn trauma, as well as acting as a regulator of certain “hub” genes in a model of type 2 diabetes [149,150]. MiR3690 has been investigated in several studies as a potential biomarker in colorectal and thyroid cancers, as well as hepatocellular carcinoma recurrence following liver transplant [151,152,153].

MiR6089 can be divided into two functional groups: inflammation and tumorigenesis. This microRNA has been found to play a key role in rheumatoid arthritis through innate-receptor signaling and macrophage-related inflammation [154,155,156,157]. MiR6089 in peripheral blood plasma has also been found to be a potential biomarker of retinoblastoma [158]. MiR12120 has been found to actively target SARS-CoV-2 during infection [159].

### 3.3. circRNA from the Y Chromosome

Circular RNAs (circRNAs) are a unique class of non-coding RNAs. They exist as single-stranded RNAs that form a covalently closed loop [160]. These looped RNAs are generated by back-splicing and are extremely stable, having a much longer half-life than mRNA [161]. The lack of a 5′ cap and a 3′ poly (A) tail further adds to the uniqueness of these RNAs. While the existence of mammalian circular RNA transcripts has been known since the 1990s, investigations of their potential functions were delayed due to the belief that they were the result of splicing errors [61,162]. However, advances in sequencing technology allowed for a better appreciation of circRNAs, as they were found to be abundant across different cell types [161,163]. Further contributing to the increased appreciation of circRNAs was emerging evidence of their role in regulating gene expression. Some circRNAs act as miRNA sponges, sequestering miRNA and upregulating or downregulating their target gene’s expression [164,165]. They further regulate gene expression by encoding peptides, changing transcription and binding protein complexes [166,167,168].

Evidence of circRNAs originating from the Y chromosome has existed for several decades, with Capel et al. detailing circular transcripts of the murine Sry gene in 1993 [169]. However, the authors noted that no circular transcript of Sry was observed in humans. While Y-linked circRNAs have been extensively described in other mammals, only a small number of circRNAs originating from the human Y chromosome have been identified and the function of these circRNAs remains largely unknown, e.g., the CIRCpedia v2 circRNA database, which contains more than 180 RNA-seq datasets, predicts the expression of 309 circRNAs from 36 coding and non-coding genes present in the human Y chromosome, including circ-ZFY, circ-USP9Y, circ-DDX3Y, circ-KDM5D, and circ-VAMP7 [170]. However, this prediction has not had significant investigation, as most of these circRNAs have not been characterized. This discrepancy between species could be due to differences in Y chromosome structure. As previously described, the human Y chromosome is notoriously covered in heterochromatin, making identification and translation of genes difficult. This may be why the mouse Y chromosome has had significantly more Y chromosome-linked circRNAs ascribed to it. Xu et al. showed that the human Y chromosome contains the fewest circRNAs, and attributed the lack of circRNAs to their short length and number of genes [171]. In a study of sex-chromosome aneuploidy effects on circRNA expression, Johannsen et al. identified 54 circRNAs from the Y chromosome [172].

Although Y-linked circRNAs role in human diseases remains largely unexplored, recent reports have demonstrated their potential as biomarkers. Luan et al. described a urinary exosomal circRNA ChrY:15,478,147–15,481,229 that is associated with more severe renal dysfunction in males compared to females with IgA nephropathy [173]. Further, several Y-linked circRNAs are upregulated in coronary heart disease [174]. Many circRNAs are suggested to originate from the Y chromosome, but there is significantly more work that needs to be done, now that the full sequence of the Y chromosome has been released through the T2T consortium. These circRNAs have the potential to explain certain sex disparities in cancer, autoimmune diseases, and microbial infections.

## 4. Discussion

The Y chromosome has long been assumed to be mostly biologically inert, beyond the 55 or so proteins it encodes. However, males and females at large have experienced different rates of diseases, both physically and mentally. This difference has been ascribed to genetic, hormonal, and environmental differences; however, the genetic differences are mainly attributed to the work of the SRY gene. Additionally, loss of the Y chromosome is associated with many diseases, including heart failure and cancer, but the reason for its loss is currently unknown. mLOY has been studied within cancer, and originally, it was assumed the Y chromosome was lost because it primarily held no use for the cell. However, further studies have identified a host of non-coding RNAs from the Y chromosome. A summary of the noncoding RNAs identified in this review is shown in Figure 2.

NcRNAs offer a new insight into potential differences between males and females. The Y chromosome, for all its challenges, offers a new template to be used for identifying novel, functional non-coding RNAs. With the release of the full telomere–telomere sequence, cloning advances, and sequencing techniques, studying ncRNAs such as lncRNAs, miRNAs and circRNAs coming from the Y chromosome has incredible potential. lncRNAs are the most prolific family of ncRNAs identified from the Y chromosome, with varied functions and applications. Not all lncRNAs have been found to be implicated in disease either, with a few found to be important biomarkers for treatments and good prognostic indicators. MiRNAs and circRNAs both have been identified on the Y chromosome, with most having a co-expressor on the X chromosome. MiRNAs have been identified as regulated in trauma response and cancer, with their expression regulating critical genes. circRNAs have been found in normal tissue expression, with very few identified to originate from the Y chromosome.

The findings presented here suggest that the Y chromosome has fewer characterized ncRNAs compared to other chromosomes. However, this does not mean there are not others present that have not been validated. circRNAs offer a strong contender for the most prevalent ncRNA on the Y chromosome, with over 300 identified, but very few characterized. Further research into epigenetic and epitranscriptomic changes, such as new molecular classification, such as epitranscriptomics (m6A, m5C, m1A, deamination, pseudourylation), and glycoRNAs, could also elucidate new mechanisms for function or regulation over these non-coding RNAs and their potential [175,176]. The lncRNAs identified present significant opportunity for therapeutic benefit, with many identified to be prognostic or biomarkers for disease. In the context of personalized medicine, there is ample opportunity to treat patients based on the presence of the Y chromosome. For example, if a male patient presents with esophageal carcinoma, targeting linc00278 may decrease the patient’s cancer progression. Additionally, the radio-sensitizing RNA group linc-SPRY3 might be a marker to identify if a male patient will have a stronger reaction to radiotherapeutic treatment. However, identifying if a patient has a Y chromosome is easier than trying to identify if a patient has a specific RNA or gene. The personalized medicine approach could even be extended to a much simpler approach, assessing Y chromosome presence in patients, as opposed to a specific RNA or gene.

Overall, this review highlights the current research on the non-coding RNA field, specifically lncRNAs, microRNAs, and circular RNAs, related to the Y chromosome. Little is known about the miRNAs and circRNAs from the Y chromosome, and with the discovery of the full Y chromosome sequences, the possibility for novel ncRNAs to be identified leaves the story of the Y chromosome at a biological cliffhanger, just waiting to be read.

## Figures and Tables

**Figure 1 ncrna-10-00021-f001:**
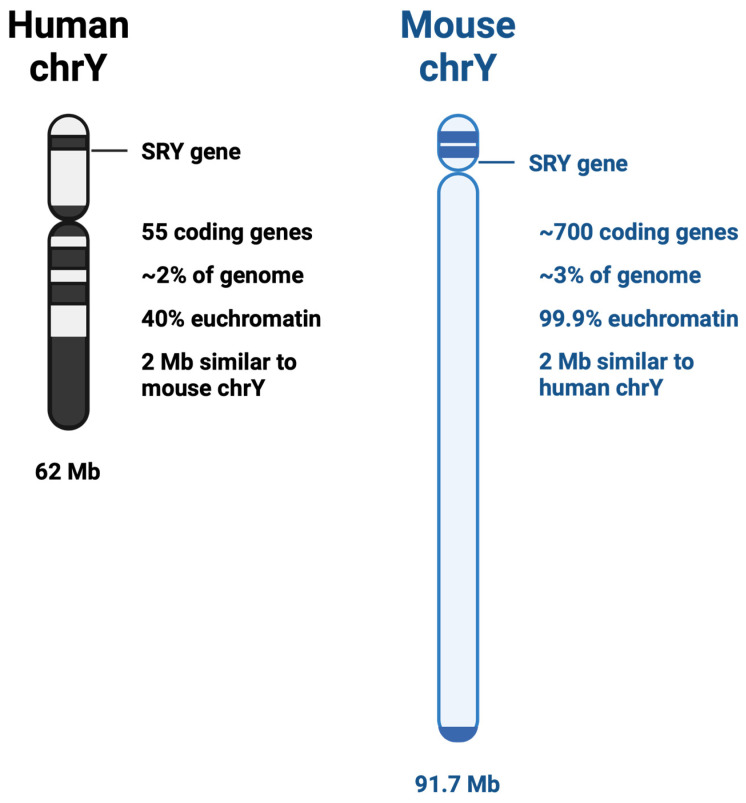
Comparison between human and mouse Y chromosomes.

**Figure 2 ncrna-10-00021-f002:**
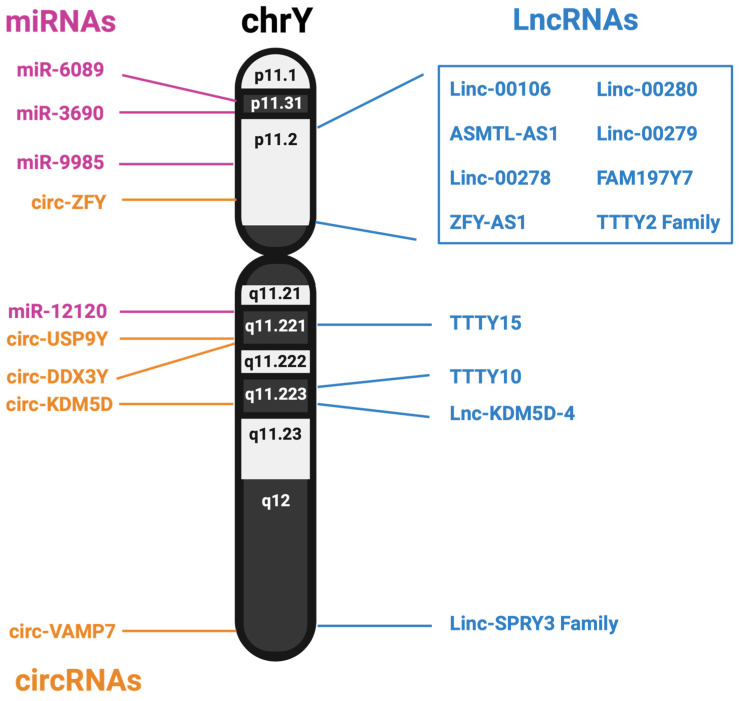
Genomic location of lncRNAs, miRNAs and predicted circRNAs expressed from the human Y chromosome.

**Table 1 ncrna-10-00021-t001:** lncRNAs originating from the Y chromosome.

Name of lncRNA	Genomic Location (GRCh38)	Function of Disease State Impact	Citations
Linc00106	chrY:1,397,025–1,399,412 chrX:1,392,420–1,401,611	Shown to sponge Let7f in hepatocellular carcinoma, increasing metastatic phenotypes	Liang et al., 2021 [107]
ASMTL-AS1	chrY:1,401,769–1,403,493 chrX:1,400,531–1,415,421	Shown to increase aggression of hepatocellular carcinoma after radiation treatment	Ma et al., 2020 [106]
linc00278	chrY:2,918,373–3,590,925	encodes Ying Yang 1 binding micropeptide, known to contribute to esophageal carcinoma	Wu et al., 2020 [101]
ZFY-AS1	chrY:2,965,356–3,002,929	Bioinformatic analysis identified as a protective biomarker in multiple myeloma	Zhou et al., 2015 [100]
Linc00280	chrY:6,357,218–6,369,921	Associated with chronic headaches when methylated	Winsvold, et al., 2018 [99]
Linc00279	chrY:8,550,518–8,713,825	Identified as a competitive endogenous RNA sponging has-mir485-5p in pulmonary tuberculosis patients	Li et al., 2022 [108]
FAM197Y7	chrY:9,367,802–9,377,092	influences bone metabolism in coronary artery calcification	Wicik, et al., 2021 [98]
TTTY2 Family	ChrY: 9,721,669–9,758,630	Deletion of the TTTY2 region results in spermatogenesis problems and early developmental issues	Yapijakis et al., 2015 [96]
TTTY15	chrY:12,537,650–12,860,839	miRNA sponge in gastric cancer, esophageal squamous cell carcinoma, and prostate cancer	Wen et al., 2022, Wang and Yang 2020, Xiao et al., 2019 [103,104,105]
TTTY10	chrY:20,375,319–20,824,330	Suggested master regulator of over 350 genes in colorectal cancer progression, primarily in cell adhesion and differentiation	Zhu et al., 2017 [102]
lnc-KDM5D-4	chrY: 20,519,948–20,524,433	Potential transcription factor in coronary artery disease	Molina, et al., 2017 [97]
linc-SPRY3 family	chrY:56,675,832–56,678,566, chrY:56,703,707–56,707,491, chrY:56,748,794–56,752,370	Suggested biomarker for radiation sensitivity in non-small-cell lung cancer	Brownmiller et al., 2020 [81]

**Table 2 ncrna-10-00021-t002:** MiRNAs originating from the Y chromosome.

Name of miRNA	Genomic Location (GRCh38)	Function of Disease State Impact	Citations
MiR9985	chrY:4,606,120–4,606,228	Differentially expressed after burn trauma and type 2 diabetes	Song et al. [149] Yin et al. [150]
MiR3690	chrY:1,293,918–1,293,992	Colorectal and thyroid cancer prognostic marker and differentially expressed in hepatocellular carcinoma	Zhang et al. [151] Shen et al. [152] Huang et al. [153]
MiR6089	chrY:2,609,191–2,609,254	Contributes to arthritis progression by increasing inflammation and acts as a prognostic marker for retinoblastoma	Yang et al. [154] Donghua et al. [155] Suxian et al. [156] Yan et al. [157] Li et al. [158]
MiR12120	chrY:13,479,177–13,479,266	SARS-CoV-2 targeting miRNA	Fulzele et al. [159]

## Data Availability

This review did not create new data.

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
