# Peer review of "More than the SRY: The Non-Coding Landscape of the Y Chromosome and Its Importance in Human Disease"

_ncrna, 2024, doi:10.3390/ncrna10020021_

Round 1

Reviewer 1 Report

Comments and Suggestions for Authors

The manuscript “More than the SRY: The Non-coding Landscape of the Y Chromosome and Their Importance in Human Disease” contains an overview of information about the sequences of non-coding RNAs described in the composition of the human Y chromosome, as well as their potential functions and putative role in health and pathology. The description is based on published data, as well as the most modern assemblies and preprints, which significantly increases the relevance of the work. I believe that the work will be of interest to readers, and will also be important for the final destruction of the myth of the selfishness of “junk DNA”.

Specific comments:

At least in the introduction, it is necessary to mention such types of non-coding RNAs as piRNA and rasiRNA, which are found in gametogenesis.

Supplement presentation of data from sections 3.2. and 3.3 with tables similar to table 1.

Complete the review with data on transcripts of human satellite 3, which is located not only in the pericentromeric region of human Y chromosome, but is also represented by multiple copies in the q12 segment. There are published data on the transcription of this repeat element in response to stress, as well as during malignancy. Some of this data is not mapped in the genome due to the presence of this repeat in other locations, however, when describing human Y chromosome that carries the largest amount of this repeat in its composition, these data cannot be ignored.

Author Response

The authors would like to thank Reviewer 1 for their thorough review of the publication. We have responded to their comments, which added significance to the work. Our responses are detailed below:

Comment 1: At least in the introduction, it is necessary to mention such types of non-coding RNAs as piRNA and rasiRNA, which are found in gametogenesis.

Response: We absolutely agree that the mention of piRNAs and rasiRNAs are important. However, evaluation of current literature has found it difficult to identify characterized rasiRNAs in humans, and as such, we reflect that in our introduction. We further clarify that we will be focusing on lncRNA, miRNA and circRNAs throughout the review.

Comment 2: Supplement presentation of data from sections 3.2. and 3.3 with tables similar to table 1.

Response: As suggested by the reviewer, we have supplemented section 3.2 with a table similar to table 1. However, with the extent of the number of proposed circRNAs identified in CIRCpedia, and the lack of characterization, we did not add a table to section 3.3.

Comment 3: Complete the review with data on transcripts of human satellite 3, which is located not only in the pericentromeric region of human Y chromosome, but is also represented by multiple copies in the q12 segment. There are published data on the transcription of this repeat element in response to stress, as well as during malignancy. Some of this data is not mapped in the genome due to the presence of this repeat in other locations, however, when describing human Y chromosome that carries the largest amount of this repeat in its composition, these data cannot be ignored.

Response: With the importance of Human Satellite III on the Y chromosome, we clarified and added significant information on HS3 to Section 2 to give further context to the difficulties of the initial sequencing of the Y chromosome.

Reviewer 2 Report

Comments and Suggestions for Authors

Dear Authors,

I was invited to review the manuscript you submitted, titled More than the SRY: The Non-coding Landscape of the Y Chromosome and Their Importance in Human Disease. The research article has considerable strengths: it articulates a rather insightful analysis mapping out the connections between the Y chromosome and the non-coding RNA universe originating from it, with a close focus on its relation to long non-coding RNAs, microRNAs and circular RNAs.

The article is somewhat competently assembled, well-structured and has several qualities constituting its chief strengths: originality, relevance and potential appeal to a relatively wide readership; the article is also quite comprehensive as it pertains to outlining and pursuing its stated objective. The methodology appears to be sound and well-explicated, as far as I could establish. Tables and figures are also well-designed and therefore effective in conveying key aspects and findings. 

The only flaws which ought to be addressed are:

- the article's objective is not stated clearly or thoroughly enough. Shedding a light on the connection between the Y chromosome and the non-coding RNA universe and how that may affect the prevalence of disease vis-a-vis environmental factors is a clear objective, but please explain more thoroughly why and how what you set out to illustrate is significant to the research community in such an essential field as cancer research (which is extensively mentioned); besides, the methodology which has been relied on to make the review has not been adequately described. It needs far more elaboration.

- Furthermore, the Discussion in my view falls relatively short compared to the level of thoroughness and accuracy which the Authors were able to achieve while pursuing the article's chief objective. Hence, in order to highlight the significance and applicability of your findings, I strongly advise you to pursue a higher degree of contextualization and elaboration as to the study's findings. For instance, the innovative potential approaches arising from personalized/precision medicine, molecular classifications, ncRnas, which though still evolving avenues for diagnostics and prognostics, have already been harnessed in various types of neoplastic conditions. A comparative analysis would surely go a long way towards broadening the scope of the article as a whole.

Furthermore, more comprehensive analysis ought to be provided as to the "tailored approaches" that loom on the horizon also from the standpoint of updating policy-making and evidence-based guidelines, as it is already under discussion in many countries. Such elements of discussion are highly consequential for clinicians and researchers, since they are likely to have a bearing on possible malpractice claims if the most suitable therapeutic avenues are not pursued. Therefore, when elaborating on the importance of tailored approaches, a higher degree of contextualization is warranted, or that remark will stay underdeveloped.

The following sources should be drawn upon and cited:

Tang H, Wu Z, Zhang Y, Xia T, Liu D, Cai J, Ye Q. Identification and Function Analysis of a Five-Long Noncoding RNA Prognostic Signature for Endometrial Cancer Patients. DNA Cell Biol. 2019 Dec;38(12):1480-1498. doi: 10.1089/dna.2019.4944.

Cavaliere AF, Perelli F, Zaami S, Piergentili R, Mattei A, Vizzielli G, Scambia G, Straface G, Restaino S, Signore F. Towards Personalized Medicine: Non-Coding RNAs and Endometrial Cancer. Healthcare (Basel). 2021 Jul 30;9(8):965. doi: 10.3390/healthcare9080965.

Perakis SO, Thomas JE, Pichler M. Non-coding RNAs Enabling Prognostic Stratification and Prediction of Therapeutic Response in Colorectal Cancer Patients. Adv Exp Med Biol. 2016;937:183-204. doi: 10.1007/978-3-319-42059-2_10. 

Piergentili R, Basile G, Nocella C, Carnevale R, Marinelli E, Patrone R, Zaami S. Using ncRNAs as Tools in Cancer Diagnosis and Treatment-The Way towards Personalized Medicine to Improve Patients' Health. Int J Mol Sci. 2022 Aug 19;23(16):9353. doi: 10.3390/ijms23169353. 

Gulìa C, Signore F, Gaffi M, Gigli S, Votino R, Nucciotti R, Bertacca L, Zaami S, Baffa A, Santini E, Porrello A, Piergentili R. Y RNA: An Overview of Their Role as Potential Biomarkers and Molecular Targets in Human Cancers. Cancers (Basel). 2020 May 14;12(5):1238. doi: 10.3390/cancers12051238. 

The article is praiseworthy in many respects: it is well-crafted and relatively well written overall. The consequences of its findings and potential prospects and applications are underdeveloped points. With some adjustments and a broader scope, it could make for a highly meaningful contribution to a highly relevant research area and benefit a relatively broad scholarly readership.

Sincerely.

Author Response

The Authors would like to thank Reviewer 2 for their highly insightful comments on the work submitted, which added significant contextualization to the work. We have responded to all of the comments, which allowed us to improve the clarity and importance of the work. Our responses are below:

Comment 1: The article's objective is not stated clearly or thoroughly enough. Shedding a light on the connection between the Y chromosome and the non-coding RNA universe and how that may affect the prevalence of disease vis-a-vis environmental factors is a clear objective, but please explain more thoroughly why and how what you set out to illustrate is significant to the research community in such an essential field as cancer research (which is extensively mentioned); besides, the methodology which has been relied on to make the review has not been adequately described. It needs far more elaboration.

Response: The authors agree the article’s objective was not clearly stated. Our objective has been edited and clarified in the final paragraph of section 1. We have added additional comments about the methodology as well. 

Comment 2: Furthermore, the Discussion in my view falls relatively short compared to the level of thoroughness and accuracy which the Authors were able to achieve while pursuing the article's chief objective. Hence, in order to highlight the significance and applicability of your findings, I strongly advise you to pursue a higher degree of contextualization and elaboration as to the study's findings. For instance, the innovative potential approaches arising from personalized/precision medicine, molecular classifications, ncRnas, which though still evolving avenues for diagnostics and prognostics, have already been harnessed in various types of neoplastic conditions. A comparative analysis would surely go a long way towards broadening the scope of the article as a whole.

Response: Reviewer 2 is correct in the limitations of the discussion. We have added additional contextualization and comparison, including the importance of personalized medicine and the potential for additional research, particularly in circRNA research.

Comment 3: Furthermore, more comprehensive analysis ought to be provided as to the "tailored approaches" that loom on the horizon also from the standpoint of updating policy-making and evidence-based guidelines, as it is already under discussion in many countries. Such elements of discussion are highly consequential for clinicians and researchers, since they are likely to have a bearing on possible malpractice claims if the most suitable therapeutic avenues are not pursued. Therefore, when elaborating on the importance of tailored approaches, a higher degree of contextualization is warranted, or that remark will stay underdeveloped. The following sources should be drawn upon and cited:

Tang H, Wu Z, Zhang Y, Xia T, Liu D, Cai J, Ye Q. Identification and Function Analysis of a Five-Long Noncoding RNA Prognostic Signature for Endometrial Cancer Patients. DNA Cell Biol. 2019 Dec;38(12):1480-1498. doi: 10.1089/dna.2019.4944.

Cavaliere AF, Perelli F, Zaami S, Piergentili R, Mattei A, Vizzielli G, Scambia G, Straface G, Restaino S, Signore F. Towards Personalized Medicine: Non-Coding RNAs and Endometrial Cancer. Healthcare (Basel). 2021 Jul 30;9(8):965. doi: 10.3390/healthcare9080965.

Perakis SO, Thomas JE, Pichler M. Non-coding RNAs Enabling Prognostic Stratification and Prediction of Therapeutic Response in Colorectal Cancer Patients. Adv Exp Med Biol. 2016;937:183-204. doi: 10.1007/978-3-319-42059-2_10.

Piergentili R, Basile G, Nocella C, Carnevale R, Marinelli E, Patrone R, Zaami S. Using ncRNAs as Tools in Cancer Diagnosis and Treatment-The Way towards Personalized Medicine to Improve Patients' Health. Int J Mol Sci. 2022 Aug 19;23(16):9353. doi: 10.3390/ijms23169353.

Gulìa C, Signore F, Gaffi M, Gigli S, Votino R, Nucciotti R, Bertacca L, Zaami S, Baffa A, Santini E, Porrello A, Piergentili R. Y RNA: An Overview of Their Role as Potential Biomarkers and Molecular Targets in Human Cancers. Cancers (Basel). 2020 May 14;12(5):1238. doi: 10.3390/cancers12051238.

Response: The authors agree, contextualization is incredibly important to the significance of the paper, and we have added additional resources into the importance of using RNA in medicine, particularly with the success of the SARS-CoV-2 vaccines. After careful consideration and research into the suggested articles mentioned here, we have agreed on their relevance, and added them as citations [69-73], to grant additional contextualization to our review.

Round 2

Reviewer 2 Report

Comments and Suggestions for Authors

Dear Authors,

I can certainly appreciate the extent to which you have succeeded in improving your manuscript. The fundamental objective, which is now more comprehensively laid out, is of great relevance and interest to a relatively broad scholarly readership.

All the additions you have made have greatly contributed to rendering the article more thorough and above all, well-balanced in its presentation.

Given all such aspects, I am recommending the Editor approve the manuscript for publication.

Best regards.